# Resolving dual binding conformations of cellulosome cohesin-dockerin complexes using single-molecule force spectroscopy

**Markus A Jobst[1,2], Lukas F Milles[1,2], Constantin Schoeler[1,2], Wolfgang Ott[1,2], Daniel B Fried[3], Edward A Bayer[4], Hermann E Gaub[1,2], Michael A Nash[1,2]***

[1]Lehrstuhl für Angewandte Physik, Ludwig-Maximilians-University, Munich, Germany; [2]Center for Nanoscience, Ludwig-Maximilians-University, Munich, Germany; [3]Kean University, New Jersey, United States; [4]Department of Biological Chemistry, The Weizmann Institute of Science, Rehovot, Israel

**Abstract** Receptor-ligand pairs are ordinarily thought to interact through a lock and key mechanism, where a unique molecular conformation is formed upon binding. Contrary to this paradigm, cellulosomal cohesin-dockerin (Coh-Doc) pairs are believed to interact through redundant dual binding modes consisting of two distinct conformations. Here, we combined site-directed mutagenesis and single-molecule force spectroscopy (SMFS) to study the unbinding of Coh:Doc complexes under force. We designed Doc mutations to knock out each binding mode, and compared their single-molecule unfolding patterns as they were dissociated from Coh using an atomic force microscope (AFM) cantilever. Although average bulk measurements were unable to resolve the differences in Doc binding modes due to the similarity of the interactions, with a single-molecule method we were able to discriminate the two modes based on distinct differences in their mechanical properties. We conclude that under native conditions wild-type Doc from *Clostridium thermocellum* exocellulase Cel48S populates both binding modes with similar probabilities. Given the vast number of Doc domains with predicted dual binding modes across multiple bacterial species, our approach opens up new possibilities for understanding assembly and catalytic properties of a broad range of multi-enzyme complexes.

*For correspondence: michael.nash@lmu.de

**Competing interests:** The authors declare that no competing interests exist.

## Introduction

Cellulosomes are hierarchically branching protein networks developed by nature for efficient deconstruction of lignocellulosic biomass. These enzyme complexes incorporate catalytic domains, carbohydrate binding modules (CBMs), cohesin:dockerin (Coh:Doc) pairs, and other conserved features (*Demain et al., 2005*; *Bayer et al., 2004*; *Schwarz, 2001*; *Béguin and Aubert, 1994*; *Smith and Bayer, 2013*; *Fontes and Gilbert, 2010*). A central attribute of cellulosome assembly is the conserved ~75 amino acid type-I Doc domain typically found at the C-terminus of cellulosomal catalytic domains. The highly conserved consensus Doc sequence from *Clostridium thermocellum (Ct)* is shown in *Figure 1A*. Dockerins guide attachment of enzymes into the networks by binding strongly to conserved Coh domains organized within non-catalytic poly (Coh) scaffolds. In addition to their nanomolar binding affinities, many archetypal Coh:Doc pairs are thought to exhibit dual binding modes (*Carvalho et al., 2007*; *Pinheiro et al., 2008*; *Currie et al., 2012*). The bound Doc domain can adopt two possible orientations that differ by ~180° rotation on the Coh surface, as shown in *Figure 1B*. The two binding modes originate from duplicated F-hand sequence motifs, a conserved structural feature found among type-I dockerins (*Pagès et al., 1997*). The duplicated F-hand motifs resemble EF-hands found in eukaryotic calcium binding proteins (e.g., calmodulin), and provide

**eLife digest** Some bacteria use cellulose, the main component of plant cell walls, as a food source. The enzymes that break down cellulose are anchored onto a protein scaffold in a structure called the cellulosome on the bacteria's surface. This anchoring occurs through an interaction between receptor proteins known as 'cohesin' domains on the scaffold proteins and 'dockerin' ligands on the enzymes.

Most receptor-ligand interactions only allow the two proteins to bind in a single, fixed orientation. However, cohesins and dockerins are suspected to bind in two different configurations. It has been difficult to investigate the populations of these different configurations because most experimental techniques investigating protein binding take average measurements from many molecules at once. As the binding modes are extremely similar, these methods have been unable to distinguish between the two cohesin-dockerin binding configurations without introducing mutations, in part because these configurations are very similar to each other.

Jobst et al. used a technique called single-molecule force spectroscopy to investigate cohesin-dockerin interactions between individual molecules. This technique applies a force that separates, or 'unbinds', cohesin and dockerin, by pulling individual complexes of the two binding partners apart with a nanoscale probe. In the experiments, *E. coli* bacteria were made to produce mutant versions of dockerin that can only bind to cohesin in one orientation. This allowed each binding configuration to be studied individually. The results of these experiments revealed the mechanical unbinding patterns of each cohesin-dockerin configuration, and showed that it is possible to use these patterns to distinguish between the two configurations. A complimentary set of experiments revealed that wild-type (non-mutated) cohesin-dockerin complexes occupy both configurations in approximately equal amounts, and do not switch modes once bound.

Further single-molecule experiments together with computer simulations will provide a more detailed picture of how cohesin and dockerin fit together in the two configurations. Such experiments could also reveal how cohesin and dockerin contribute to the break down of cellulose inside living cells and how they could be used for the precise assembly of single proteins.

internal sequence and structural symmetry to Doc domains. Rotating Doc by ~180° with respect to Coh (*Figure 1B,C*) results in an alternatively bound complex with similarly high affinity involving the same residues on Coh recognizing mirrored residues within Doc. The dual binding mode is thought to increase the conformational space available to densely packed enzymes on protein scaffolds, and to facilitate substrate recognition by catalytic domains within cellulosomal networks (*Bayer et al., 2004*). From an evolutionary perspective, the dual binding mode confers robustness against loss-of-function mutations, while allowing mutations within Doc to explore inter-bacterial species cohesin-binding promiscuity in cellulosome-producing microbial communities. Coh:Doc interactions and dual binding modes are therefore important in the context of cellulose degradation by cellulosome-producing anaerobic bacterial communities.

However, direct experimental observation of the dual binding modes for wild-type Doc has thus far proven challenging. Ensemble average bulk biochemical assays (e.g., surface plasmon resonance, calorimetry, enzyme-linked immunosorbent assays) are of limited use in resolving binding mode populations, particularly when the binding modes are of equal thermodynamic affinity. Crystallography is challenging because the complex does not adopt a unique molecular conformation, but rather exhibits a mixture of two conformations thereby hindering crystal growth. Structural data on the dual binding mode have typically been collected using a mutagenesis approach, where one of the binding modes was destabilized by mutating key recognition elements (*Carvalho et al., 2007*; *Pinheiro et al., 2008*). This approach, however, while resolving the structures of each bound complex, cannot determine if one binding mode is dominant for wild-type Doc, or if that dominance is species or sequence dependent. Coarse grained molecular dynamics has also predicted dual modes of interaction between Coh and Doc (*Hall and Sansom, 2009*), but direct experimental evidence of both binding modes for wild-type Doc has remained elusive. Improved fundamental understanding of the dual binding mode could shed light onto the molecular mechanisms by which these multi-

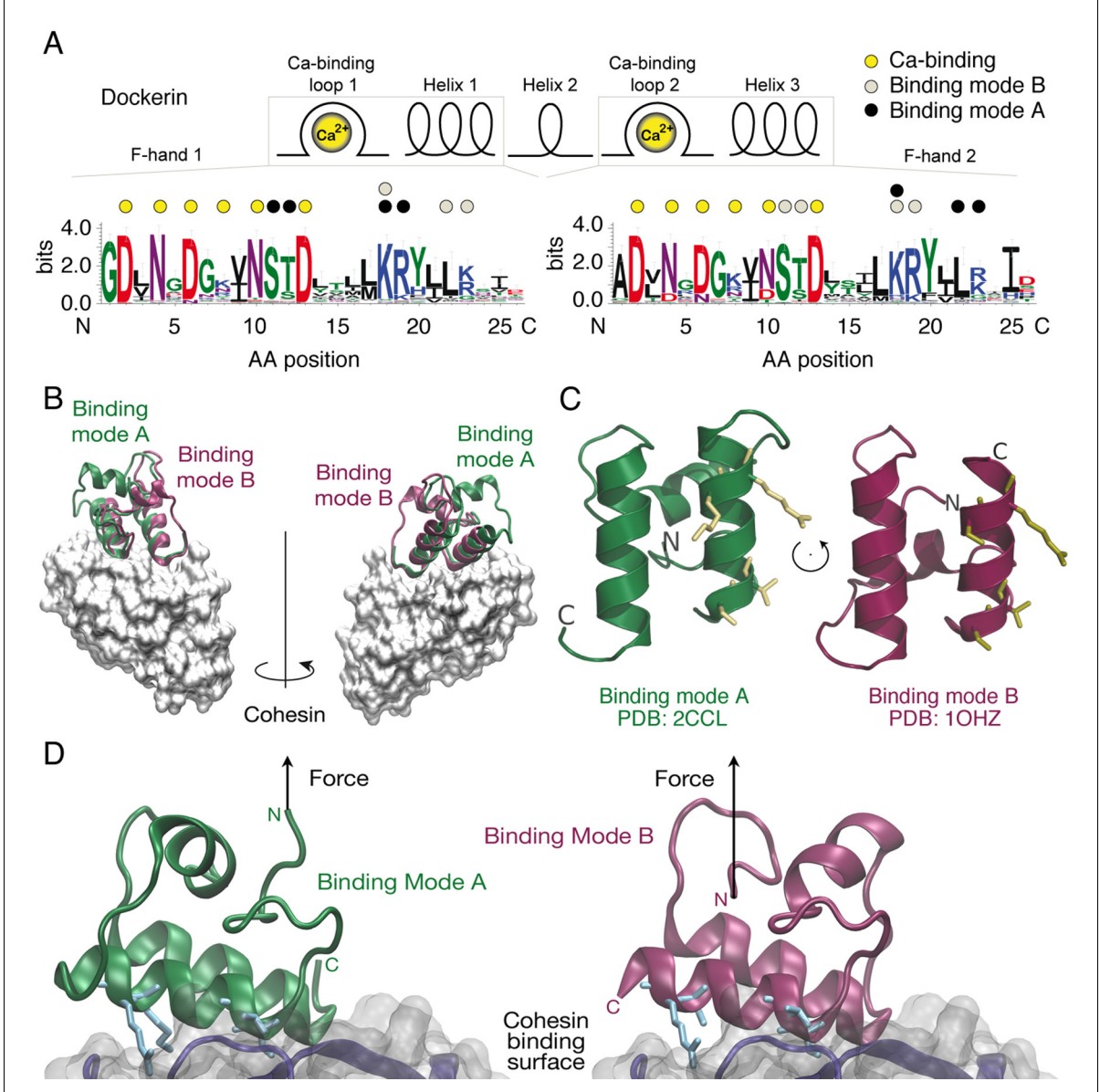

**Figure 1.** Cohesin:Dockerin dual binding modes. (**A**) Secondary structure and consensus sequence logo (**Crooks, 2004**) assembled from 65 putative *Ct* type-I Doc variants. Dots above the amino acid codes indicate residues involved in: $Ca^{2+}$ coordination (yellow), mode A binding (black), and mode B binding (gray). Letter colors represent chemical properties: Green, polar; purple, neutral; blue, basic; red, acidic; black, hydrophobic. Crucial Coh-binding residues are located at positions 11, 12, 18, 19, 22, and 23 in each F-hand motif. (**B**) Coh:Doc complex crystal structures showing overlaid Doc domains in the two binding modes. Images were generated by aligning the Coh domain (gray) from PDB 2CCL (green, binding mode (**A**) and 1OHZ (red, binding mode (**B**) using the VMD plugin MultiSeq (**Humphrey et al., 1996**; **Roberts et al., 2006**). (**C**) View of the Doc binding interface for each mode from the perspective of Coh. The conserved binding residues at positions 11, 12, 18, and 19 in the F-hand motif relevant for binding in the corresponding mode are depicted as stick models (yellow). (**D**) Close-up view of the interface for each binding mode with arrows indicating the location and direction of applied force. Binding residues 11, 12, 18, and 19 for binding mode A and 45, 46, 52, and 53 for binding mode B are shown as blue stick models. The Coh domain is oriented the exact same way in both views.

enzyme complexes self-assemble and achieve synergistic conformations, as well as provide a new approach to designing systems for protein nanoassembly (**Kufer et al., 2009**; **2008**).

Here, we used SMFS (**Li and Cao, 2010**; **Engel and Müller, 2000**; **Woodside and Block, 2014**) to study wild-type and mutant Doc from exocellulase Cel48S of *C. thermocellum* (*Ct*-DocS). We demonstrate that specific unfolding/unbinding trajectories of individually bound Coh:Doc complexes

are characteristic of the binding modes. To validate our approach, we produced Doc mutants that exhibited a preferred binding mode. We performed single-molecule pulling experiments on bound Coh:mutant Doc complexes and observed a strong bias in the probability of two clearly distinguishable unfolding patterns, termed 'single' and 'double' rupture types for each binding mode mutant. We further probed the unbinding mechanism of the double rupture events using poly (Gly-Ser) inserts to add amino acid sequence length to specific sections of Doc as a means to identify which portions of Doc unfolded. Finally, we used the inherent differences in mechanical stability of each binding mode, and the effects these differences had on the unfolding force distributions of an adjacent domain, to directly observe and quantify binding mode populations for wild-type Doc.

## Results

### Protein design

The wild-type and mutant Doc sequences used in this work were aligned (*Beitz, 2000*) and are presented in *Figure 2*. Among *Ct*-Doc domains, a Ser-Thr pair located at positions 11 and 12 of F-hand motif 1 (N-terminal helix 1) is highly conserved (*Figure 1A*). This Ser-Thr pair is H-bonded to Coh in binding mode A (*Figure 1A*, black dots). Analogously, binding mode B refers to the configuration where the Ser-Thr pair from helix 3 dominates the H-bonding to Coh (*Figure 1A*, gray dots). Binding mode B was previously crystallized for a homologous *Ct*-Doc (*Carvalho et al., 2003*). Mutation of the Ser-Thr pair in helix 3 to Ala-Ala was used to bias binding and thereby crystallize binding mode A for the same Doc (*Carvalho et al., 2007*). A similar targeted mutagenesis approach was also used to obtain crystal structures of a *Clostridium cellulolyticum* Doc in each binding mode (*Pinheiro et al., 2008*).

To preferentially select for a specific binding mode (A or B), we prepared Doc sequences that incorporated 4 amino acid point mutations, referred to as quadruple mutants ('Q'). To design quadruple mutants, we noted that recent structural work reported a set of *Ct*-Doc domains that differ from the canonical duplicated Ser-Thr sequences. These non-canonical Docs were found to exhibit only a single binding mode (*Brás et al., 2012*; *Pinheiro et al., 2009*). In one of these non-canonical Doc domains, an Asp-Glu pair was found in place of Ser-Thr. Since the Coh surface is negatively charged, we postulated that including Asp-Glu in place of Ser-Thr within one of the F-hands could be used to effectively knock out a given binding mode for our canonical Doc. Additionally, we incorporated double alanine mutations to replace the conserved Lys-18 Arg-19 residues of a given F-hand motif, further destabilizing a targeted binding mode. Q1 refers to a quadruple mutant where helix 1 has been modified at four positions (i.e. S11D-T12E-K18A-R19A). Q3 refers to the quadruple mutant where helix 3 has been modified at four positions (i.e. S43D-T44E-K50A-R51A). As a negative

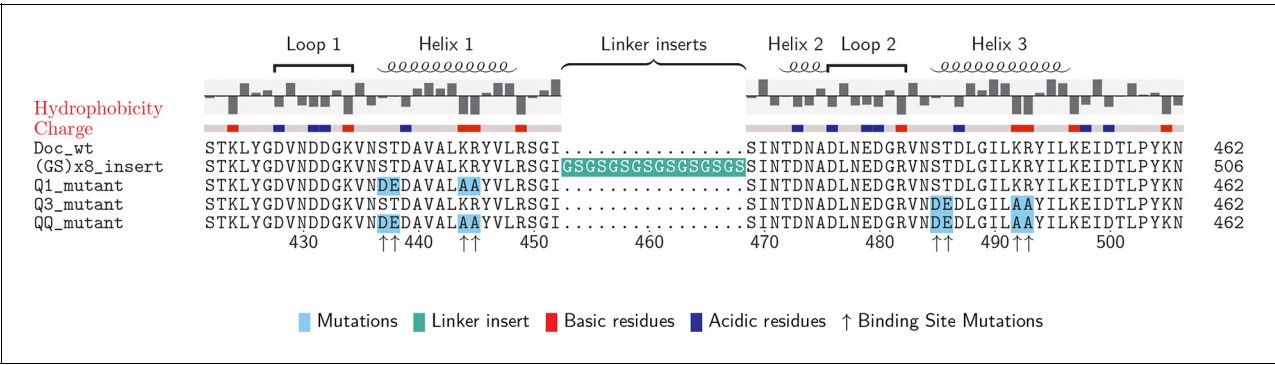

**Figure 2.** Doc sequences used in this study (N- to C-terminus). Doc_wt: wild-type sequence; hydrophobicity and charge graphs are displayed for the wild-type-Doc (red: positively charged, blue: negatively charged); (GS)x8_insert: A (Gly-Ser)8 linker was incorporated between helix 1 and helix 2; Q1_mutant: Quadruple mutant in helix 1. Four point mutations (DE/AA) were incorporated into Doc helix 1 to knock out binding mode A; Q3_mutant: Quadruple mutant in helix 3. Four point mutations (DE/AA) were incorporated into Doc helix 3 to knock out binding mode B; QQ_mutant: Non-binding control with both binding modes knocked out. Numbers below indicate amino acid number of the fusion protein construct starting from the xylanase N-terminus.

control, we prepared a mutant referred to as 'QQ' that incorporated quadruple mutations into both helices 1 and 3.

Doc domains were expressed as fusion domains attached to the C-terminal end of xylanaseT6 (Xyn) from *Geobacillus stearothermophilus* to improve solubility and expression levels as previously reported (*Stahl et al., 2012*). The Xyn domain also acts as a so-called fingerprint in AFM force extension traces to provide a means for screening datasets and searching for known contour length increments. We use the term 'contour length' to refer to the maximum length of a stretched (unfolded) polypeptide chain. Our screening process identified single-molecule interactions and ensured correct pulling geometry. For the Coh domain, we chose cohesin 2 from *Ct*-CipA expressed as a C-terminal fusion domain with the family 3a carbohydrate binding module (CBM) from *Ct*-CipA. In order to exclude artifacts arising from fingerprint domains, protein immobilization or pulling geometry, a second set of fusion proteins was cloned, expressed and probed in complementary experiments using a flavoprotein domain from the plant blue light receptor phototropin (iLOV) (*Chapman et al., 2008*). All protein sequences are provided in the 'Materials and methods' section.

## Single-molecule unfolding patterns

The pulling configuration for single-molecule AFM experiments is shown in *Figure 3A*. CBM-Coh was site-specifically and covalently attached to an AFM cantilever tip and brought into contact with a glass surface modified with Xyn-Doc. The mechanical strength of protein domains and complexes will strongly depend on the pulling points (i.e. sites at which the molecule is attached to cantilever/ surface). The site-specific attachment chemistry used here was precisely defined by the chosen residue of immobilization, ensuring the same loading geometry was used on the complex for each and every data trace. After formation of the Coh:Doc complex, the cantilever was retracted at a constant speed that ranged from 200 to 3200 nm/s while the force was monitored by optical cantilever deflection. The resulting force-distance traces were characteristic of the series of energy barriers crossed by the protein complex along the unfolding/unbinding pathway. A sawtooth pattern was consistently observed when molecular ligand-receptor complexes had formed. Sorting the data using contour length transformation (*Puchner et al., 2008*) and identifying traces that contained a Xyn contour length increment (~89 nm) allowed us to screen for single-molecule interactions (*Stahl et al., 2012*), as described in our prior work on Coh:Doc dissociation under force (*Stahl et al., 2012; Schoeler et al., 2014; Jobst et al., 2013; Otten et al., 2014; Schoeler et al., 2015*).

Typical single-molecule interaction traces from such an experiment are shown in *Figure 3B, C* and in *Figure 3—figure supplement 1*. Following PEG linker stretching, an initial set of peaks

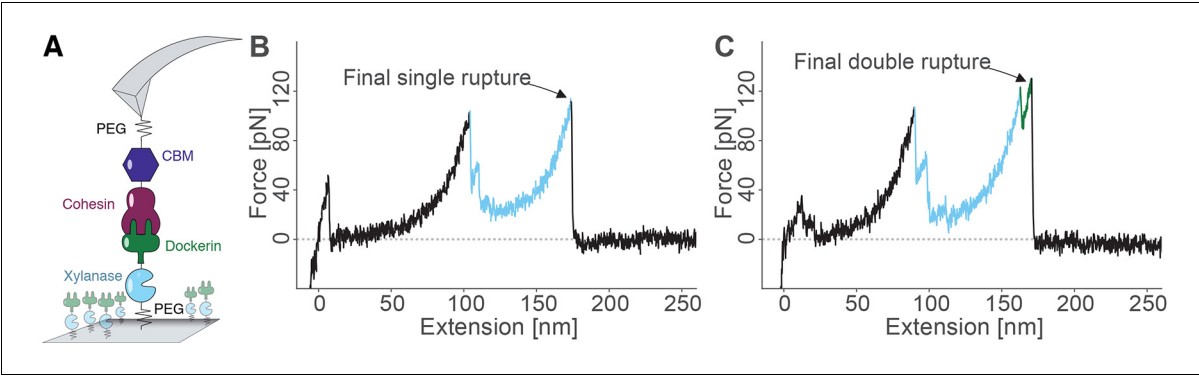

**Figure 3.** Overview of the experimental configuration and recorded single-molecule unfolding and unbinding traces. (A) Schematic depiction showing the pulling geometry with CBM-Coh on the AFM Cantilever and Xyn-Doc on the glass substrate. Each fusion protein is site-specifically and covalently immobilized on a PEG-coated surface. (B-C) Each force vs. extension trace shows PEG linker stretching (black), xylanase unfolding and subsequent stretching (blue), and Coh:Doc complex rupture. The Coh:Doc complex rupture occurred in two distinct event types: single (B) and double (C) ruptures. The 8-nm contour length increment separating the double peaks was assigned to Doc unfolding (C, green).

The following figure supplement is available for figure 3:

**Figure supplement 1.** Representative sample of force traces.

sequentially decreasing in force was assigned to xylanase unfolding and stretching. This domain when unfolded added ~89 nm of free contour length to the system. The final peak (s) corresponded to rupture of the Coh:Doc complex, and occurred as either 'single' or 'double' rupture events. The contour length increment between the two double event peaks was found to be ~8 nm, that is, 8 nm of hidden contour length was added to the biopolymer during a sub-step of Doc unbinding (see 'Discussion'). The 8-nm contour length increment was also observed in complementary experiments employing other fusion domains: xylanase was swapped for an sfGFP domain and CBM was swapped out for an iLOV domain. In these new fusions, the 8 nm Doc increment was still observed, indicating it was not caused by a specific fusion domain. As we show below, double and single rupture events were associated with binding modes A and B, respectively. CBM unfolding length increments (~57 nm) were only rarely observed because the Coh:Doc complex only rarely withstood forces sufficiently high to unfold CBM (*Stahl et al., 2012*).

## Ensemble average binding experiments

Binding experiments were carried out in bulk to evaluate the binding affinity of wild-type, Q1, Q3, and QQ Doc sequences to wild-type Coh. Xyn-Doc fusion protein variants were immobilized in a microwell plate and exposed to tag red fluorescent protein (TagRFP) (*Merzlyak et al., 2007*) fused to Coh (TagRFP-Coh) across a range of concentrations, followed by rinsing and subsequent fluorescence readout (*Figure 4A*). The data clearly showed that Q1 and Q3 Doc sequences, each with a mutated binding mode, maintained high-binding affinity with dissociation constants ($K_d$) in the nM range. These values are in good agreement with previous reports on homologous type-I Doc domains (*Brás et al., 2012*; *Sakka et al., 2011*). This suggested that mutant Doc domains with one destabilized binding mode were still able to recognize fluorescent protein fused Coh with strong affinity by relying on the alternative binding mode that was preserved. The QQ double knockout mutant, however, showed no appreciable binding over the concentration range tested. This negative control showed that DEAA quadruple mutations were in fact effective at eliminating binding for the targeted modes.

## Single-molecule rupture statistics of binding mode mutants

For each Doc tested, we collected tens of thousands of force-extension traces and selected for further analysis only those traces showing the ~89 nm xylanase contour length increments and no other anomalous behavior, resulting in typically 200–3000 usable single-molecule interaction curves per experiment. We determined the number of Coh:Doc unbinding events that occurred as single or double rupture peaks. The results are shown in *Figure 4B*. The wild-type Doc showed double rupture events in ~57% of the cases, and single rupture events in ~43% of the cases. The mutant designed to knock out binding mode A (Q1), showed a single event probability of ~77%, and a double event probability of ~23%. The mutant designed to knock out binding mode B (Q3) showed a single event probability of ~41%, and a double event probability of ~59%. It is clear from these data that the Q1 mutant has a strong bias toward single peaks that is not observed in the wild-type leading to preliminary assignment of single peaks to binding mode B.

For all double events, we determined the force difference of the second peak relative to the first (*Figure 4C*). Q1 and wild-type on average showed second peaks that were ~15–20% higher in force than the first peak. Q3 meanwhile showed clearly different behavior. Although the ratios of single to double peaks were nearly identical between wild-type and Q3, differences in the relative force between the first and second peaks differentiated wild-type and Q3 (*Figure 4C*). Double peaks for the Q3 mutant were more likely to show a shielded behavior, where the second peak was lower in force than the first peak by ~10%. Although the Q3 mutant showed the same single vs. double event probability as wild-type, the double events for Q3 were distinguishable from those of the wild-type based on this observed decrease in the rupture force of the second peak. The second barrier of the double events was therefore weaker in Q3 than for wild-type. This weaker 2nd double peak for the Q3 mutant combined with similar single/double peak ratios as wild-type leads us to believe that the number of double peaks is being underestimated systematically for the Q3 mutant. Generally, each binding mode still allows for the occurrence of a single event (albeit with different likelihood), in which the whole Doc domain unbinds without an additional unfolding substep. Since the second and final energy barrier for complex dissociation is weaker than the first for the Q3 mutant, the

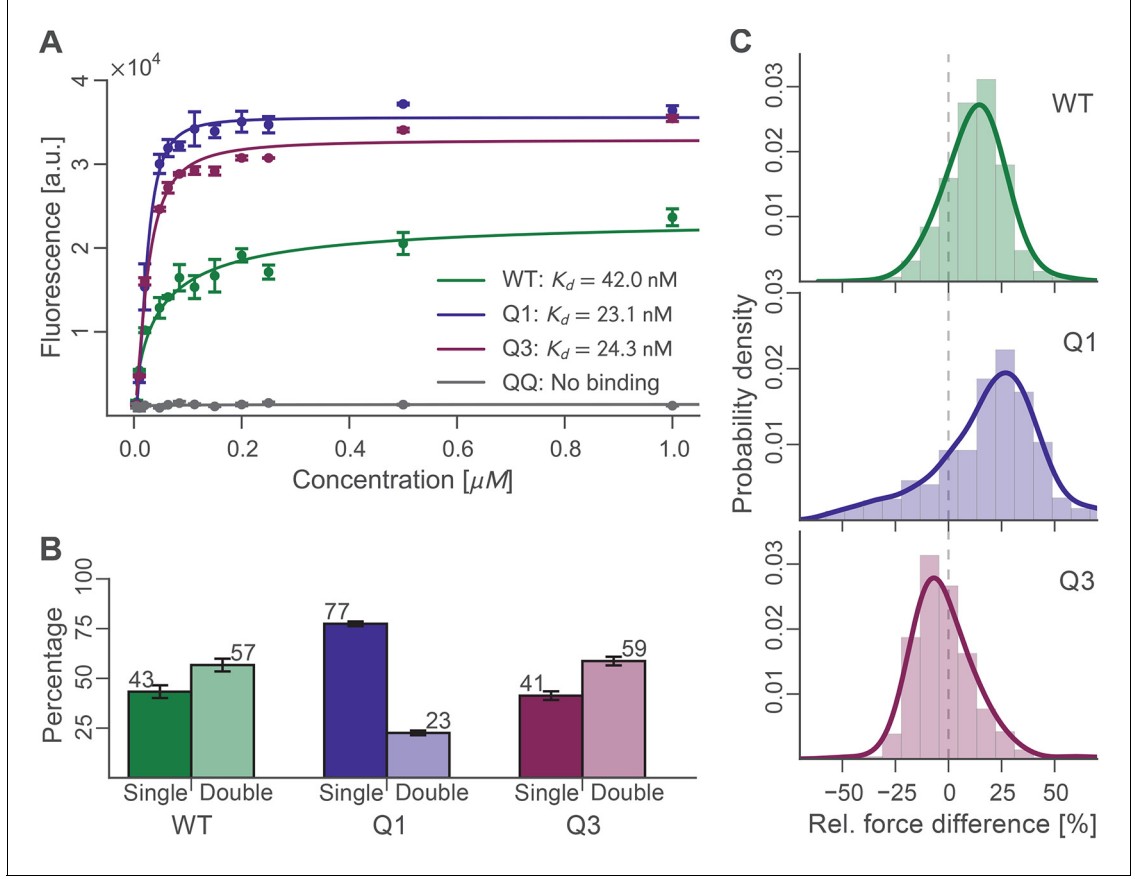

**Figure 4.** Bulk and single-molecule characterization of Doc mutants. (**A**) Fluorescence binding curve showing binding of TagRFP-labelled Coh to wild-type and mutant Doc nonspecifically immobilized in a 96-well plate. Both Q1 and Q3 mutants bound TagRFP-Coh similarly to wild-type with dissociation constants ($K_D$) in the low nM range. The negative control QQ mutant showed no binding. Solid lines are 4 parameter logistic nonlinear regression model fits to the data. Error bars represent the standard deviation of three independent samples. (**B**) Event probabilities for single (opaque colors) and double (translucent colors) Coh:Doc rupture peaks determined for Doc wild-type and DE/AA quadruple mutants. Data originate from 947, 4959, and 1998 force-extension traces from wild-type, Q1 and Q3 variants, respectively. Error bars represent 95% Clopper-Pearson confidence intervals based on the beta probability distribution. (**C**) Relative difference in double peak rupture forces for the different Doc variants. Positive values indicate a stronger final peak. Histograms represent concatenated data from various pulling speeds. Drawn lines are kernel density estimates calculated on the raw data.

The following source data is available for figure 4:

**Source data 1.** Probability Data.

probability for the molecule to pass both barriers simultaneously is increased, thus resulting in a higher percentage of single events.

## Probing the 8-nm length increment with poly (GS) inserts

We sought to identify the molecular origin of the 8 nm contour length increment separating the double event peaks by engineering additional amino acid sequence length into the Doc domain. Amino acid insert sequences have previously been used to probe length increments in AFM force spectroscopy experiments (*Bertz and Rief, 2009*) (*Carrion-Vazquez et al., 1999*). By adding additional amino acids to the polypeptide chain at a particular location, insert sequences increase the gain in contour length following unfolding of a subdomain in a predictable way. Any change in the observed length increment can be pinpointed to the position in the molecule where the unfolding event occurs. In this case, we engineered flexible $(GS)_8$ insert sequences directly into wild-type Doc between helices 1 and 2, in a flexible loop that was not expected to interfere with either of the two binding modes. Structural homology models (*Figure 5A*) of the wild-type Doc and $(GS)_8$ insert

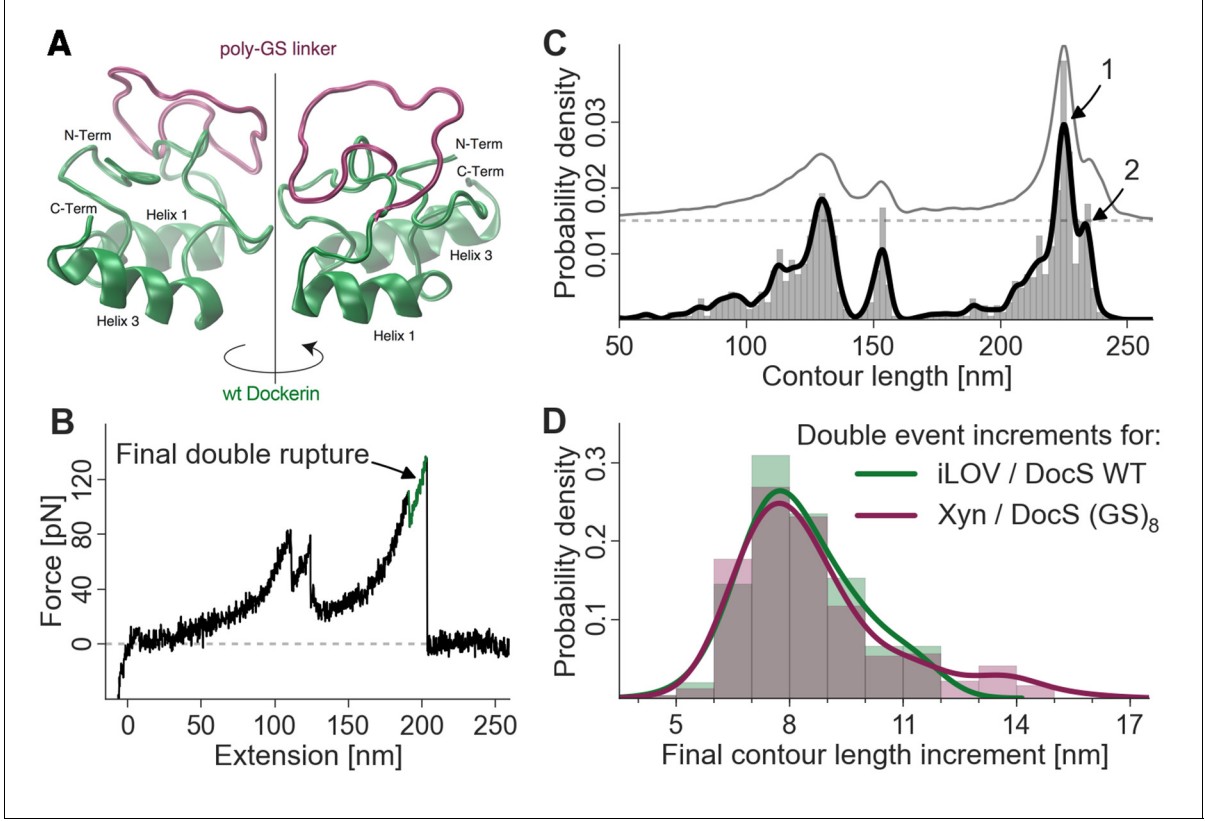

**Figure 5.** Probing the final contour length increment with Poly (GS) inserts. (**A**) Structural homology model overlay of wild-type and mutant Doc containing a $(GS)_8$-linker between helix 1 and helix 3. The wild-type Doc is shown in green. The 16 amino acid long GS-insert is shown in purple (*Kelley and Sternberg, 2009*) (remaining Doc domain not shown). (**B**) Typical force extension trace with final double rupture event depicted in green (arrow). (**C**) Histogram and kernel density estimate of the transformation of the single force extension trace in panel B into contour length space (black) and kernel density estimate of the whole dataset of single molecule Xyn-Doc:Coh-CBM traces bearing xylanase fingerprint and final double rupture (gray, offset in y-direction for readability) in contour length space. (**D**) Histograms (bars, bin width: 1 nm), kernel density estimates (drawn lines, bandwidth: 0.75 nm, gaussian kernel), and statistical test (Kolmogorov-Smirnov, 'KS test') are each calculated on the raw data of the final increments (peak-to-peak distances) in contour length space (x-distance between arrow 1 and 2 in panel (**C**). Maxima for final double event increments lie at 7.75 nm and 7.73 nm for iLOV-Coh:Doc (wild-type)-sfGFP (N = 255) and Xyn-Doc $(GS)_8$:Coh-CBM (N = 320) final ruptures, respectively (a two-sample KS test on the raw data indicates no significant difference in the data distributions (p-value of 21.7%).

sequence were calculated using the Phyre server (*Kelley and Sternberg, 2009*). If the 8-nm contour length increment was caused by sequential unbinding of Doc helices 1 and 3 in wild-type Doc, then double peaks for the poly (GS) constructs should show an increase in the double peak contour length increment. As shown in *Figure 5B,C and D*, the contour length histogram for $(GS)_8$ Doc was indistinguishable from the wild-type Doc. No additional contour length was gained due to additional amino acids inserted between Doc helices 1 and 2. Since the Doc was anchored to the glass slide through an N-terminal xylanase domain, this result indicated that the unfolding event responsible for the 8-nm length increment must be located upstream (i.e. N-terminal) from the site of the $(GS)_8$-insert. This result suggested that unfolding of calcium binding loop 1 and helix 1 in Doc was the source of the 8-nm length increment.

## Single-molecule evidence of dual binding mode

To finally confirm the presence of both bound conformations in wild-type Coh:Doc complexes, we replaced xylanase with sfGFPand CBM with iLOV as the contour length marker or fingerprint domains. iLOV was chosen as a superior unfolding fingerprint domain because it does not show multiple unfolding substeps (in contrast to xylanase), which simplified analysis. Also iLOV has an unfolding force distribution that lies in a similar range as the Coh:Doc complex dissociation single and

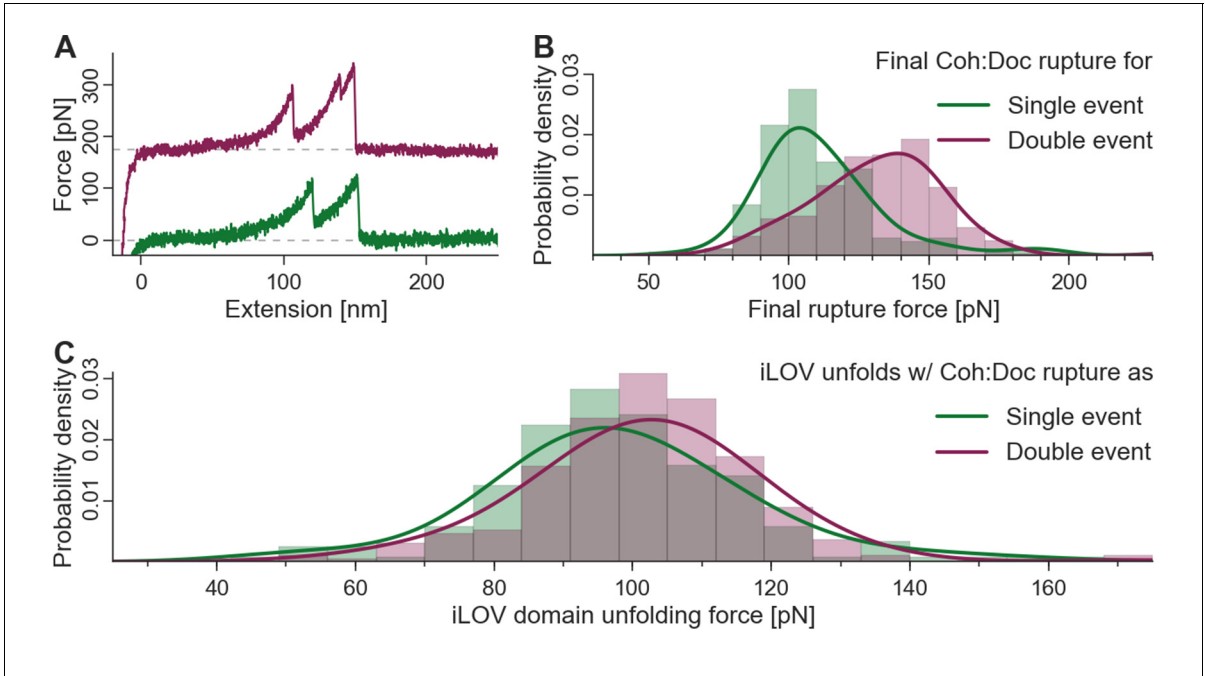

**Figure 6.** Biasing of unfolding force distributions by dual binding mode. (**A**) Typical force traces showing iLOV unfolding with final single (green) and double (purple) complex ruptures. The curve terminating in a double peak is offset in the y-direction for clarity. (**B**) Final complex rupture force distribution for single and double events. Double events are more mechanically stable. (**C**) iLOV domain unfolding forces for final single (green) and double (red) events at a pulling velocity of 800 nm/s. Histograms (bars), kernel density estimates (lines), and statistical tests are each obtained from the raw data. Maxima for iLOV unfolding lie at 96.0 pN and 102.7 pN for single (N = 172) and double (N = 277) final ruptures, respectively. A two-sample Kolmogorov-Smirnov test showed significant differences in the data distributions (p-value of 0.09%). Since the data were all recorded with a single cantilever and both event types were distributed equally throughout the runtime of the measurement, no systematic biasing is expected. Because of the lower force distribution of final single peaks, the iLOV unfolding force distribution is truncated compared to final double peak force traces, supporting the notion that the binding mode is set prior to mechanical loading of the complex.

double peaks, allowing for effective biasing of the iLOV unfolding force distributions by the inherent stability difference between single and double event peaks. *Figure 6A* shows characteristic single and double event curves containing iLOV unfolding (36-nm contour length increment) followed by Coh:Doc rupture as a single or double event. The rupture force distributions of the single and double event (second peak) ruptures are shown in *Figure 6B*. The most probable rupture force for single events was ~104 pN, while for double events this value was ~140 pN at a pulling speed of 800 nm/s. We next calculated the unfolding force distributions of the iLOV domain for curves that terminated with single events or double events. If the Coh:Doc complex ruptured before iLOV unfolding was observed, the curve was eliminated from the dataset because it lacked a fingerprint domain length increment. This criterion for inclusion in the dataset results in a biasing of the iLOV unfolding forces, since the maximum of the fingerprint unfolding force distribution that can be observed must lie below that of the Coh:Doc complex. The fact that we observed a downward shift in the iLOV unfolding forces (*Figure 6C*) for curves that terminated in the less mechanically stable single rupture event is confirmation that the single- and double-event peaks arise from separate bound conformations. Each mode has a distinct mechanical stability and energy landscape that is set at the time of receptor-ligand binding, that is once bound, the conformation of the complex does not change. If single- and double-event unbinding patterns were simply two competing pathways out of the same bound state, then the downward shift in rupture force distribution would not be observed for the iLOV unfolding forces. Although this shift in rupture force distributions is comparatively subtle, it can be observed accurately with high statistical significance. We note that the datasets for both binding modes were measured with the same cantilever throughout the runtime of the whole experiment. Calibration and drift issues therefore did not interfere with the required accuracy.

## Discussion

The relatively small ~8 kDa Doc domains exhibit an internal sequence and structural symmetry that is believed to give rise to a dual mode of binding to Coh, as shown in *Figure 1*. In order to study this remarkable plasticity in molecular recognition in greater detail, we prepared a series of mutants (*Figure 2*) designed to either knock out a specific binding mode or add length to the molecule at a specific position. Bulk experiments showed that Doc mutants Q1 and Q3, originally designed to suppress one of the binding modes, were still able to bind Coh with high affinity, while the double knockout did not bind (*Figure 4A*). The equilibrium affinities of Coh binding to Q1, Q3, or wild-type were all similarly high with $K_D$s in the low nM range, in good agreement with literature values (*Sakka et al., 2011*), suggesting the two binding modes are thermodynamically equivalent and rendering them indistinguishable with conventional methods such as ELISA or calorimetry. Techniques like surface plasmon resonance could possibly show differing values for on- and off-rates for the mutants, but would still not be able to resolve the binding modes within a wild-type population.

Force spectroscopy with the AFM interrogates individual molecules, and measures their mechanical response to applied force. Since the technique is able to probe individual members of an ensemble, it provided a means to quantify binding mode configurations by assigning unfolding/unbinding patterns to the binding mode adopted by the individual complexes. Site-directed Q1 and Q3 mutations supported the assignment of binding mode A to a characteristic double rupture peak dissociation pathway. Single events were assigned to binding mode B and showed no Doc unfolding substep prior to complex rupture.

We consistently observed 8 nm of added contour length that separated the Doc double peaks. Since force is applied to Doc from the N-terminus, we analyzed the Doc sequence starting at the N-terminus and searched for reasonable portions of Doc that could unfold in a coordinated fashion to provide 8 nm of contour length. The results from the GS-insert experiments (*Figure 5*) indicated no change in the double-event contour length increment, regardless of the added GS-insert length located between helix 1 and 3 in Doc. This result is consistent with the 8 nm length increment being located N-terminally from the GS-insert site, implicating unfolding of Doc calcium binding loop 1 and helix 1 as the source of the 8 nm. This length accurately matches the estimated length increment for unfolding calculated from the crystal structure (*Figure 1D*).

Although this result could also be consistent with the 8 nm increment being located somewhere outside the Doc domain in the polyprotein, we deem this scenario highly unlikely. The 8 nm increment cannot be located in the Xyn or CBM domains because we have accounted for Xyn and CBM lengths in their entirety based on the observed 89 nm and rare 57 nm length increments here and in a previous study (*Stahl et al., 2012*), and for confirmation swapped out those domains for different proteins completely (i.e. iLOV and GFP). The remaining possibility that the 8 nm is located within the Coh domain is also not likely since the barrel-like structure of the Coh is known to be mechanically highly stable (*Valbuena et al., 2009*; *Hoffmann et al., 2013*). Also, if the 8-nm length increment were due to partial Coh unfolding, the Q1 and Q3 mutants would not be expected to affect the single/double peak ratio or force differences between the double event peaks as was observed (*Figure 4B, C*). The GS-insert data suggest the 8-nm length increment is located within Doc, upstream (N-terminal) from the GS-insert site implicating calcium loop 1 and helix 1 in this unfolding event.

Finally, we observed that an inherent difference in the mechanical stability of single and double event rupture peaks (*Figure 6B*) could be used as a feature by which to discriminate the binding modes. Our analysis algorithm accepted only the force curves that first showed iLOV fingerprint domain unfolding followed by either a single- or double-rupture peak. By observing a small but significant downward shift in the iLOV unfolding force distribution when analyzing curves that terminated in the less stable single-event peak, we confirmed the single-event peaks originate from a unique conformation that is 'set' at the time of complex formation.

Taken together, we propose an unbinding mechanism where the first barrier of the double peaks represents unfolding of the N-terminal calcium binding loop and unraveling of alpha helix 1 up to the Lys-Arg pair at sequence positions 18 and 19 in the wild-type structure in binding mode A. Based on a length per stretched amino acid of 0.4 nm, the expected contour length for unfolding the Doc domain up to this position would be 7.6 nm, in good agreement with the measured value of 8 nm within experimental error. A portion of the N-terminal calcium binding loop (i.e. residues S11-

T12) is involved in binding to D39 in Coh. The first peak of the double events is attributed to breakage of this interaction and simultaneous unfolding of calcium loop 1 and alpha helix 1 up to the Lys-Arg pair at sequence positions 18 and 19. Another contributing factor is the intramolecular clasp that has been identified as a stabilizing mechanism among similar type-I Doc domains (*Slutzki et al., 2013*). A recent NMR structural study (*Chen et al., 2014*) on the same wild-type Doc used in this work confirmed a hydrophobic ring-stacking interaction between Tyr-5 and Pro-66. Confirmation of this clasp motif by NMR means the head and tail of the Doc are bound together, additionally stabilizing the barrier that is overcome in the first of the double event peaks. In this scenario, subsequent to breaking the interactions between the calcium binding loop and Coh, disrupting the intramolecular clasp and unfolding the N-terminal loop-helix motif, the remaining bound residues including Lys-18, Arg-19, Lys-50, Leu-54, and Lys-55 stay bound to Coh and are able to withstand substantial force on their own, eventually breaking in the second and final of the double rupture peaks. Prior work further supports this unbinding mechanism, revealing that a progressive N-terminal truncation of Doc did not affect the interaction largely, unless the truncation reached the Lys-18 and Arg-19 residues (*Karpol et al., 2009*). This corroborates the idea of the C-terminal end of helix 1 being a crucial part of the binding site within the complex. Single rupture peaks were thus observed when the wild-type complex was bound in binding mode B, and no unfolding of Ca-binding loop 1 or helix 1 occurred. Force was propagated directly to bound residues Lys-18, Leu-22, and Arg-23 which when broken resulted in complete complex dissociation.

Given the fingerprint biasing phenomenon (*Figure 6C*), we finally sought to correct the single/double peak counting statistics (*Figure 4B*) in order to correct for undercounting of single peaks due solely to their failure to reach sufficiently high forces to unfold the fingerprint domain. Only traces showing a fingerprint were analyzed to ensure defined unfolding geometry. Using the rupture force distributions of singles, doubles, iLOV, and xylanase domains, we calculated the probability of occurrence of fingerprint unfolding at a force higher than the single-event ruptures. This overlap probability was found to be 0.85 for iLOV and 0.40 for xylanase. When the single/double peak ratios for were corrected for this effect, the final binding mode ratios (binding mode A/binding mode B, i. e., doubles/singles) were found to be 0.95 and 0.87 for xylanase-Doc and iLOV, respectively. These ratios are close to 1 indicating comparable probability of each binding mode after accounting for biasing the single/double peak counting statistics due to fingerprint domain stability. We note that these numbers are also slightly lower than unity due to the exclusion of double peaks that occurred before unfolding of the fingerprint domains. Further details on rupture force distributions and overlap statistics are shown in *Figure 7*. As the magnitude of biasing changes with the unfolding force distributions of each fingerprint domain, overlaps in the probability distributions allow for normalizing single/double event ratios of experimental data sets with different fingerprinting domains. For the Coh:Doc complex unbinding event, biasing (undercounting) is more pronounced for the mechanically weaker single ruptures. This normalization procedure shows the relative difference of biasing between single and double events, as double events are less biased than single events.

The biological significance of Coh-Doc interactions in the context of cellulosome assembly and catalysis cannot be overstated. Their high affinity and specificity, along with their modularity, thermostability, and their ultrastable mechanical properties all make Coh-Doc unique from a biophysics perspective, and attractive from an engineering standpoint. Dual binding mode Doc domains are broadly predicted among many cellulosome producing bacteria (e.g. *C. thermocellum, C. cellulolyticum, R. flavefaciens*), however relatively few have been confirmed experimentally (*Carvalho et al., 2007*; *Pinheiro et al., 2008*; *Brás et al., 2012*). In fact, the direct effect of single vs. dual binding modes on the ability of cellulosomes to convert substrate into sugars is currently unknown. It is therefore unclear whether or not dual binding modes affect, for example, the catalytic properties of native or engineered synthetic cellulosomes.

However, it is important to note that cellulosome producing bacteria invariably live among communities with other microorganisms, which may be producing cellulases and cellulosomes of their own. In such an environment, a dual binding mode could enable organisms to produce enzymes that are able to bind to a neighboring species' scaffoldins, yet still retain high-affinity interactions with host scaffoldins. They would be able to combine resources with neighboring cells in a mixed microbial consortium. The dual binding mode could therefore allow genetic drift to explore interspecies protein binding. Indeed, cross-species reactivity between Coh and Doc has been reported (*Haimovitz et al., 2008*). Cellulosome-producing microbes may therefore be pursuing a middle

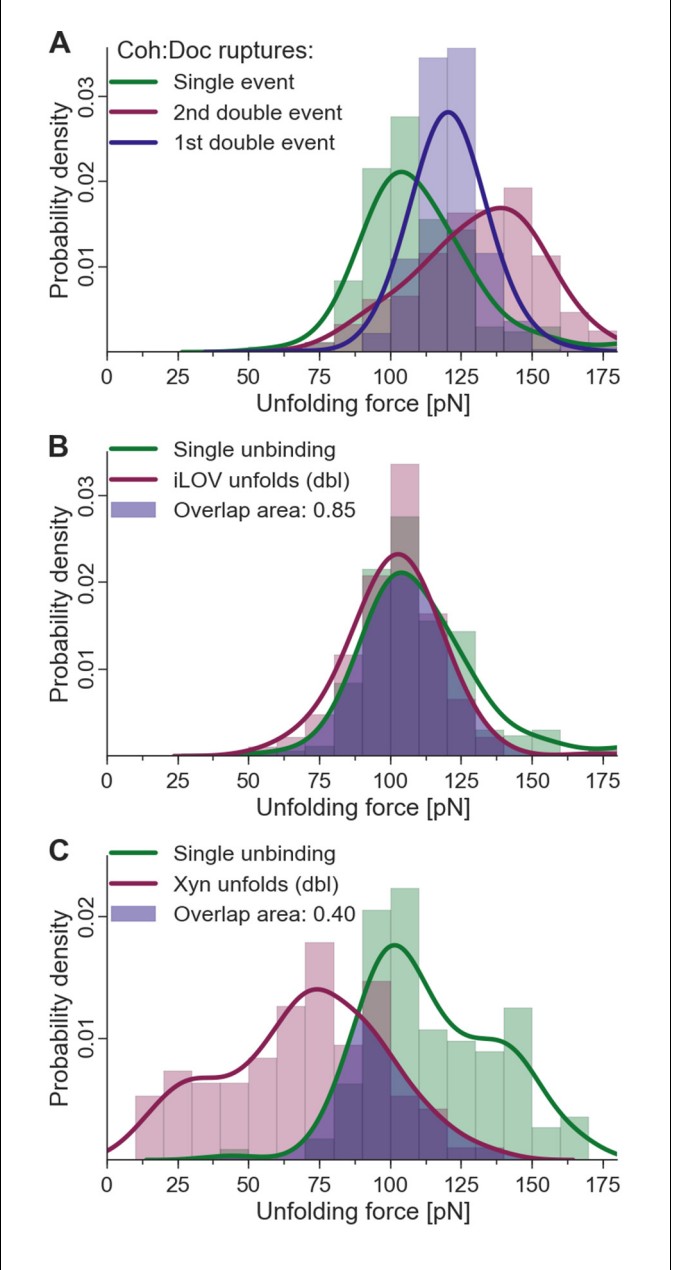

**Figure 7.** Fingerprint unfolding and complex unbinding forces. (**A**) Rupture force distribution of final complex ruptures for single (green), first (purple) and second (red) double unbinding events. (**B**) Overlap area (purple) of iLOV domain unfolding force distribution (red) (iLOV-doubles curve class) with the rupture force distribution (green) for single-event complex ruptures. (**C**) Overlap area (purple) of Xyn domain unfolding force distribution (red) (Xyn-doubles curve class) with the rupture force distribution (green) for single-event complex ruptures. Overlaps in probability distributions allow normalizing single-event counts to double events to account for different biasing caused by the different unfolding forces of the fingerprint domain. Biasing occurs, because for overlapping force distributions of fingerprint unfolding and complex ruptures, unbinding events are more likely to take place without fingerprint unfolding if the two distributions are closer together. For the Coh:Doc unbinding, this effect is more pronounced for the weaker single ruptures. Because double events are also biased, this still does not give a true quantification, but only compensates for the differences of biasing. The non-bell-evans-like shape of the single rupture peaks, especially in the region of the 1st double event peak (**A**) suggests that this class of curves does not contain a single type of unbinding mechanism, but rather a superposition of different event types.

ground between protein synthesis strictly for selfish vs. communal usage. By distinguishing the presence of each binding mode for wild-type Doc domains, the single-molecule biophysical approach presented here based on differences in mechanical hierarchies will facilitate further study into the significance of the dual binding mode.

In summary, the dual binding mode of Coh:Doc domains has so far proven resistant to explicit experimental characterization. Crystallography combined with mutagenesis has provided snapshots of the two modes, but resolving each of the modes for wild-type Doc under near native conditions has up until now not been possible. We have demonstrated the advantages of a single-molecule approach in resolving these subtle differences in molecular conformations of bound complexes. Despite having equal thermodynamic binding affinity, when mechanically dissociated by pulling from the N-terminus of Doc, binding mode A was more mechanically stable with an additional energy barrier. This mechanical difference was exploited to probe the two binding modes independently from one another, providing direct observation of this unique mechanism in molecular recognition. In the future, harnessing control over binding modes could offer new approaches to designing molecular assembly systems that achieve defined protein orientations.

## Materials and methods

### Site-directed mutagenesis of plasmid DNA

A pET28a vector containing the previously cloned xylanaseT6 from *Geobacillus stearothermophilus* (*Salama-Alber et al., 2013*) and DocS dockerin from *Clostridium thermocellum* Cel48S were subjected to QuikChange mutagenesis (*Wang and Malcolm, 1999*) to install the following mutations: Q1, Q3, and QQ in the dockerin and T129C in the xylanase, respectively.

For insertion of the $(GS)_4$ and $(GS)_8$ linkers into the Doc domain, exponential amplification with primers bearing coding sequences for the inserts at their 5'-ends was performed with a Phusion High-Fidelity DNA polymerase (New England Biolabs, MA). PCR products were then blunt end ligated using KLD Enzyme Mix and KLD Reaction Buffer from the Q5 site directed mutagenesis kit (New England Biolabs, MA). The modified DNA constructs were used to transform *Escherichia coli* DH5-alpha cells, grown on kanamycin-containing agar plates and subsequently screened. All mutagenesis products were confirmed by DNA sequencing analysis.

Primers used for inserting the $(GS)_8$ linker into the Doc domain:

Fw 5'-ggttctggctccggttctggctccagcatcaacactgacaat-3'
Rev 5'-agaaccggagccagagccggaacctatacctgatctcaaaacatatct-3'

### Protein expression and purification

Fusion proteins HIS-CBM A2C-Coh2 (*C.t.*) were expressed in *E. coli* BL21(DE3)RIPL cells in kanamycin-containing media supplemented with 2mM calcium chloride overnight at 16°C. After harvesting, cells were lysed by sonication, and the lysate was subjected to heat treatment at 60°C for 30 min to precipitate the bulk of the host bacterial proteins, leaving the expressed thermophilic proteins in solution. The lysate was then pelleted, and the supernatant fluids were applied to a beaded cellulose column and incubated at 4°C for 1 hr. The column was then washed with 50 mM Tris buffer (pH 7.4) containing 1.15 M NaCl, and the protein was eluted using a 1% (vl/v) triethylamine aqueous solution. Tris buffer was added to the eluent and the solution was neutralized with HCl.

Fusion proteins HIS-Xyn T129C-DocS (*C.t.*) wild-type, Q1, and Q3 mutants were expressed as described above. Following heat treatment, the supernatant fluids were applied to a Ni-NTA column and washed with TBS buffer containing 20mM imidazole and 2mM calcium chloride. The bound protein was eluted using TBS buffer containing 250 mM imidazole and 2 mM calcium chloride. The solution was then dialyzed to remove the imidazole.

Fusion proteins ybbR-HIS-CBM A2C-Coh2 (*C.t.*), ybbR-HIS-Xyn T129C-DocS (*C.t.*) wild-type and QQ mutants and ybbR-HIS-Xyn T129C-DocS (*C.t.*) $(GS)_4$ insert were expressed in *E. coli* BL21(DE3) RIPL cells; ybbR-HIS-Xyn T129C-DocS (*C.t.*) $(GS)_8$ insert fusion protein variants were expressed in *E. coli* NiCo21(DE3)RIPL cells. Cultivation and expression was done in ZYM-5052 autoinduction media (*Studier, 2005*) containing kanamycin (and chloramphenicol, in case of the NiCo21(DE3)RIPL cells) overnight at 22°C, overall 24 hr. After harvesting, cells were lysed using sonication. The lysate was then pelleted by centrifugation at 39,000 rcf, the supernatant fluids were applied to Ni-NTA columns

and washed with TBS buffer. The bound protein was eluted using TBS buffer containing 200 mM imidazole. Imidazole was removed with polyacrylamide gravity flow columns or with polyacrylamide spin desalting columns.

All protein solutions were concentrated with Amicon centrifugal filter devices and stored in 50% (v/v) glycerol at -20°C (ybbR-free constructs) or -80°C (ybbR-bearing constructs). The concentrations of the protein stock solutions were determined to be in the order of 1–15 mg/mL by absorption spectrophotometry at a wavelength of 280 nm.

## ELISA-like binding assay

1 µM of Xyn-Doc fusion proteins (wild-type Q1, Q3, QQ Doc fusions) bearing either wild-type or mutant Doc domains were adsorbed onto surfaces of the wells of a 96-well nunc maxi sorp plate (Thermo Scientific, Pittsburgh, PA). After blocking (2% (w/v) BSA, 0.05% Tween 20 in TBS buffer) and several rinsing steps, a red fluorescent protein-cohesin (StrepII-TagRFP-Coh2 (*C.t.*), Addgene ID 58,710 (*Otten et al., 2014*)) fusion construct was incubated to the unspecifically immobilized Doc fusion proteins over a range of concentrations. After further rinsing, the fluorescence of the TagRFP domain was measured with a multi-well fluorescence plate reader ( M1000 PRO, Tecan Group Ltd., Männedorf, Switzerland). Fluorescence values were plotted against their corresponding concentration values for each protein variant, and 4 parameter logistic nonlinear regression model functions were fitted to the data to determine the transition point of the curve.

## Surface immobilization strategies

The Xyn domain had a cysteine point mutation at position 129 (Xyn T129C) to facilitate covalent attachment to a glass surface *via* Polyethylene glycol (PEG)-maleimide linkers. There were no other cysteines within the Xyn or Doc domains, which ensured site-specific immobilization of the molecule and defined mechanical loading of Doc from the N-terminus for the AFM experiments. The CBM domain likewise contained an A2C cysteine point mutation for covalent attachment to the cantilever tip *via* PEG-maleimide linkers. The second set of fusion proteins sfGFP-Doc and iLOV-Coh was covalently attached to coenzyme A bearing PEG linkers by their terminal ybbR tags.

## AFM sample preparation

For AFM measurements, silicon nitride cantilevers (Biolever mini, BL-AC40TS-C2, Olympus Corporation nominal spring constant: 100 pN/nm; 25 kHz resonance frequency in water), and glass coverslips (Menzel Gläser, Braunschweig, Germany; diameter 22mm) were used. 3-Aminopropyl dimethyl ethoxysilane (APDMES, ABCR GmbH, Karlsruhe, Germany), α-Maleinimidohexanoic-ω-NHS PEG (NHS-PEG-Mal, Rapp Polymere, Tübingen, Germany; PEG-MW: 5 kDa), immobilized tris (2-carboxylethyl)phosphine (TCEP) disulfide reducing gel (Thermo Scientific, Pittsburgh, PA), tris (hydroxymethyl) aminomethane (TRIS, >99% p.a., Carl Roth, Karlsruhe, Germany), $CaCl_2$ (>99% p.a., Carl Roth, Karlsruhe, Germany), sodium borate (>99.8% p.a., Carl Roth, Karlsruhe, Germany), NaCl (>99.5% p.a., Carl Roth, Karlsruhe, Germany), ethanol (>99% p.a.), toluene (>99.5% p.a., Carl Roth, Karlsruhe, Germany) were used as received. Sodium borate buffer was 150 mM, pH 8.5. Measurement buffer for AFM-SMFS was tris-buffered saline supplemented with 1 mM $CaCl_2$ (TBS, 25 mM TRIS, 75 mM NaCl, 1 mM $CaCl_2$ pH 7.2). All buffers were filtered through a sterile 0.2 µm polyethersulfone membrane filter (Nalgene, Rochester, NY) prior to use.

Force spectroscopy measurement samples, measurements and data analysis were prepared and performed according to previously published protocols (*Jobst et al., 2013*;*Otten et al., 2014*). In brief, NHS-PEG-Maleimide linkers were covalently attached to cleaned and amino-silanized silicon nitride AFM cantilevers and cover glasses. The respective protein constructs were covalently linked either *via* engineered cysteine residues to the maleimide groups of the surface on the sample directly, or *via* Sfp phosphopantetheinyl transferase-mediated attachment of a terminal ybbR tag to coenzyme A, which was previously attached to the maleimide groups of the surface.

## AFM-SMFS measurements

AFM data were recorded in 25 mM TRIS pH 7.2, 75 mM NaCl and 1mM $CaCl_2$ buffer solution (TBS). Retraction velocities for constant speed force spectroscopy measurements varied between 0.2 and 3.2 µm/s. Cantilever spring constants were calibrated utilizing the thermal method applying the

equipartition theorem to the one dimensionally oscillating lever (*Hutter and Bechhoefer, 1993*; *Cook et al., 2006*). Measurements were performed on custom built instruments, deploying an Asylum Research (Santa Barbara, CA, USA) MFP-3D AFM controller and Physik Instrumente (Karlsruhe, Germany) or attocube (Munich, Germany) piezo nanopositioners (*Gumpp et al., 2009*). After each measurement, the xy-stage was actuated by 100 nm to probe a new spot on the surface and measure new individual Xyn-Doc fusion molecules. Instrument control software was programmed in Igor Pro 6.3 (Wavemetrics). The retraction speed was controlled with a closed-loop feedback system running internally on the AFM controller field-programmable gate array (FPGA).

## Force-extension data analysis

Data analysis and plotting was performed in Python (Python Software Foundation. Python Language Reference, version 2.7. Available at http://www.python.org) utilizing the libraries NumPy and SciPy (*van der Walt et al., 2011*) and Matplotlib (*Hunter, 2007*).

Measured raw data were analyzed by determining the zero force value with the baseline position and applying a cantilever bending correction to the z-position. The resulting force distance traces were coarsely screened for peaks as sudden drops in force and curves with less than three peaks (such as in *Figure 3—figure supplement 1*, panel F) were excluded, as they contain no clearly identifiable signal. Force-distance traces were transformed into contour length space with the inverse worm-like-chain model (*Jobst et al., 2013*), assuming a fixed persistence length of 0.4 nm. Screening for the 89 nm xylanase, the 36nm iLOV and the final 8 nm final double rupture increment was performed by finding their corresponding local maxima in a kernel density estimate with bandwidth b = 1 nm. Thresholds in force, distance, and peak counts were applied to sort out nonspecific and multiple interactions. All curves were ultimately selected for the xylanase or iLOV fingerprint and checked manually. For the counting statistics, double peaks were detected as an increment of 8 +- 4 nm in contour length for final rupture peaks in the contour length plot, given that the curve showed one of the fingerprints. If a double peak was detected, the force difference was determined as the percentual difference between the first and the final rupture peak force.

Barrier position diagrams were assembled using optimal alignment through cross-correlation (*Puchner et al., 2008*; *Otten et al., 2014*). The numbers of points included in fitted histograms are provided in the figure captions, along with the statistical tests and significance values obtained.

## Amino acid sequences

### pET28a-HIS-XynT129C-DocS (*C.t.*) wild-type

MSHHHHHHKNADSYAKKPHISALNAPQLDQRYKNEFTIGAAVEPYQLQNEKDVQMLKRHFNSIVAENV-
MKPISIQPEEGKFNFEQADRIVKFAKANGMDIRFHTLVWHSQVPQWFFLDKEGKPMVNECDPVKREQNK-
QLLLKRLETHIKTIVERYKDDIKYWDVVNEVVGDDGKLRNSPWYQIAGIDYIKVAFQAARKYGGDNIKLYM-
NDYNTEVEPKRTALYNLVKQLKEEGVPIDGIGHQSHIQIGWPSEAEIEKTINMFAALGLDNQITELDVSM-
YGWPPRAYPTYDAIPKQKFLDQAARYDRLFKLYEKLSDKISNVTFWGIADNHTWLDSRADVYYDANGNV-
VVDPNAPYAKVEKGKGKDAPFVFGPDYKVKPAYWAIIDHKVVPGTPSTKLYGDVNDDGKVNSTDAVALK-
RYVLRSGISINTDNADLNEDGRVNSTDLGILKRYILKEIDTLPYKN

### pET28a-ybbR-HIS-XynT129C-DocS (*C.t.*) 16aa GS Insert

MGTDSLEFIASKLALEVLFQGPLQHHHHHHPWTSASKNADSYAKKPHISALNAPQLDQRYKNEFTIGAAV-
EPYQLQNEKDVQMLKRHFNSIVAENVMKPISIQPEEGKFNFEQADRIVKFAKANGMDIRFHTLVWHSQVP-
QWFFLDKEGKPMVNECDPVKREQNKQLLLKRLETHIKTIVERYKDDIKYWDVVNEVVGDDGKLRNSPWY-
QIAGIDYIKVAFQAARKYGGDNIKLYMNDYNTEVEPKRTALYNLVKQLKEEGVPIDGIGHQSHIQIGWPSE-
AEIEKTINMFAALGLDNQITELDVSMYGWPPRAYPTYDAIPKQKFLDQAARYDRLFKLYEKLSDKISNVTFW-
GIADNHTWLDSRADVYYDANGNVVVDPNAPYAKVEKGKGKDAPFVFGPDYKVKPAYWAIIDHKVVPGT-
PSTKLYGDVNDDGKVNSTDAVALKRYVLRSGIGSGSGSGSGSGSGSGSSINTDNADLNEDGRVNSTDLGI-
LKRYILKEIDTLPYKN

### pET28a-HIS-XynT129C-DocS (*C.t.*) Q1 mutant

MSHHHHHHKNADSYAKKPHISALNAPQLDQRYKNEFTIGAAVEPYQLQNEKDVQMLKRHFNSIVAENV-
MKPISIQPEEGKFNFEQADRIVKFAKANGMDIRFHTLVWHSQVPQWFFLDKEGKPMVNECDPVKREQNK-

QLLLKRLETHIKTIVERYKDDIKYWDVVNEVVGDDGKLRNSPWYQIAGIDYIKVAFQAARKYGGDNIKLYM-
NDYNTEVEPKRTALYNLVKQLKEEGVPIDGIGHQSHIQIGWPSEAEIEKTINMFAALGLDNQITELDVSM-
YGWPPRAYPTYDAIPKQKFLDQAARYDRLFKLYEKLSDKISNVTFWGIADNHTWLDSRADVYYDANGNV-
VVDPNAPYAKVEKGKGKDAPFVFGPDYKVKPAYWAIIDHKVVPGTPSTKLYGDVNDDGKVNDEDAVALA-
AYVLRSGISINTDNADLNEDGRVNSTDLGILKRYILKEIDTLPYKN

### pET28a-HIS-XynT129C-DocS (*C.t.*) Q3 mutant
MSHHHHHHKNADSYAKKPHISALNAPQLDQRYKNEFTIGAAVEPYQLQNEKDVQMLKRHFNSIVAENV-
MKPISIQPEEGKFNFEQADRIVKFAKANGMDIRFHTLVWHSQVPQWFFLDKEGKPMVNECDPVKREQNK-
QLLLKRLETHIKTIVERYKDDIKYWDVVNEVVGDDGKLRNSPWYQIAGIDYIKVAFQAARKYGGDNIKLYM-
NDYNTEVEPKRTALYNLVKQLKEEGVPIDGIGHQSHIQIGWPSEAEIEKTINMFAALGLDNQITELDVSM-
YGWPPRAYPTYDAIPKQKFLDQAARYDRLFKLYEKLSDKISNVTFWGIADNHTWLDSRADVYYDANGNV-
VVDPNAPYAKVEKGKGKDAPFVFGPDYKVKPAYWAIIDHKVVPGTPSTKLYGDVNDDGKVNSTDAVALK-
RYVLRSGISINTDNADLNEDGRVNDEDLGILAAYILKEIDTLPYKN

### pET28a-HIS-XynT129C-DocS (*C.t.*) QQ mutant
MSHHHHHHKNADSYAKKPHISALNAPQLDQRYKNEFTIGAAVEPYQLQNEKDVQMLKRHFNSIVAENV-
MKPISIQPEEGKFNFEQADRIVKFAKANGMDIRFHTLVWHSQVPQWFFLDKEGKPMVNECDPVKREQNK-
QLLLKRLETHIKTIVERYKDDIKYWDVVNEVVGDDGKLRNSPWYQIAGIDYIKVAFQAARKYGGDNIKLYM-
NDYNTEVEPKRTALYNLVKQLKEEGVPIDGIGHQSHIQIGWPSEAEIEKTINMFAALGLDNQITELDVSM-
YGWPPRAYPTYDAIPKQKFLDQAARYDRLFKLYEKLSDKISNVTFWGIADNHTWLDSRADVYYDANGNV-
VVDPNAPYAKVEKGKGKDAPFVFGPDYKVKPAYWAIIDHKVVPGTPSTKLYGDVNDDGKVNDEDAVALA-
AYVLRSGISINTDNADLNEDGRVNDEDLGILAAYILKEIDTLPYKN

### pET28a-ybbR-HIS-sfGFP-DocIS (*C.t.*)
MGTDSLEFIASKLALEVLFQGPLQHHHHHHPWTSASSKGEELFTGVVPILVELDGDVNGHKFSVRGEGEG-
DATIGKLTLKFICTTGKLPVPWPTLVTTLTYGVQCFSRYPDHMKRHDFFKSAMPEGYVQERTISFKDDGKYK-
TRAVVKFEGDTLVNRIELKGTDFKEDGNILGHKLEYNFNSHNVYITADKQKNGIKANFTVRHNVEDGSVQL-
ADHYQQNTPIGDGPVLLPDNHYLSTQTVLSKDPNEKRDHMVLHEYVNAAGITHGMDELYKKVVPGTPST-
KLYGDVNDDGKVNSTDAVALKRYVLRSGISINTDNADLNEDGRVNSTDLGILKRYILKEIDTLPYKN

### pET28a-ybbR-HIS-CBM A2C-Coh2 (*C.t.*)
MGTDSLEFIASKLALEVLFQGPLQHHHHHHPWTSASMCNTVSGNLKVEFYNSNPSDTTNSINPQFKVTNT-
GSSAIDLSKLTLRYYYTVDGQKDQTFWCDHAAIIGSNGSYNGITSNVKGTFVKMSSSTNNADTYLEISFTG-
GTLEPGAHVQIQGRFAKNDWSNYTQSNDYSFKSASQFVEWDQVTAYLNGVLVWGKEPGGSVVPSTQP-
VTTPPATTKPPATTIPPSDDPNAGSDGVVVEIGKVTGSVGTTVEIPVYFRGVPSKGIANCDFVFRYDPNVLEII-
GIDPGDIIVDPNPTKSFDTAIYPDRKIIVFLFAEDSGTGAYAITKDGVFAKIRATVKSSAPGYITFDEVGGFAD-
NDLVEQKVSFIDGGVNVGNAT

### pET28a-ybbR-HIS-iLOV-Coh2 (*C.t.*)
MGTDSLEFIASKLALEVLFQGPLQHHHHHHPWTSASGSPEFIEKNFVITDPRLPDNPIIFASDGFLELTEYSR-
EEILGRNARFLQGPETDQATVQKIRDAIRDQRETTVQLINYTKSGKKFWNLLHLQPVRDQKGELQYFIGV-
QLDGSDHVGSVVPSTQPVTTPPATTKPPATTIPPSDDPNAGSDGVVVEIGKVTGSVGTTVEIPVYFRGVPSK-
GIANCDFVFRYDPNVLEIIGIDPGDIIVDPNPTKSFDTAIYPDRKIIVFLFAEDSGTGAYAITKDGVFAKIRATV-
KSSAPGYITFDEVGGFADNDLVEQKVSFIDGGVNVGNAT

### pET28a-StrepII-TagRFP-Coh2 (*C.t.*)
MWSHPQFEKVSKGEELIKENMHMKLYMEGTVNNHHFKCTSEGEGKPYEGTQTMRIKVVEGGPLPFAFDI-
LATSFMYGSRTFINHTQGIPDFFKQSFPEGFTWERVTTYEDGGVLTATQDTSLQDGCLIYNVKIRGVNFPS-
NGPVMQKKTLGWEANTEMLYPADGGLEGRSDMALKLVGGGHLICNFKTTYRSKKPAKNLKMPGVYYVD-
HRLERIKEADKETYVEQHEVAVARYCDLPSKLGHKLNGSVVPSTQPVTTPPATTKPPATTIPPSDDPNAGSD-
GVVVEIGKVTGSVGTTVEIPVYFRGVPSKGIANCDFVFRYDPNVLEIIGIDPGDIIVDPNPTKSFDTAIYPDRKII-
VFLFAEDSGTGAYAITKDGVFAKIRATVKSSAPGYITFDEVGGFADNDLVEQKVSFIDGGVNVGNAT

## Acknowledgements

The authors acknowledge Carlos Fontes, Sarah Teichmann, Stefan Stahl, and Ellis Durner for helpful discussions. Support for this work was provided by the ERC Advanced Grant CelluFuel, and the EU 7th Framework Programme NMP4- SL-2013-604530 (CellulosomePlus), and the German-Israeli Foundation (GIF) for Scientific Research and Development. MAN acknowledges support from Society in Science – The Branco Weiss Fellowship from ETH Zurich.

## Additional information

### Funding

| Funder | Grant reference number | Author |
|---|---|---|
| European Research Council | 294438 | Hermann E Gaub |
| European Commission | NMP4- SL-2013-604530 | Daniel B Fried |
| German-Israeli Foundation for Scientific Research and Development | G-147-207.4-2012 | Edward A Bayer<br>Hermann E Gaub<br>Michael A Nash |
| Society in Science | Branco Weiss Fellowship | Michael A Nash |

The funders had no role in study design, data collection and interpretation, or the decision to submit the work for publication.

### Author contributions

MAJ, Conception and design, Acquisition of data, Analysis and interpretation of data, Drafting and revising the article, Contributed reagents; LFM, Conception and design, Analysis and interpretation of data, Drafting and revising the article, Contributed unpublished essential data or reagents; CS, Acquisition of data, Analysis and interpretation of data, Drafting and revising the article; WO, Acquisition of data, Contributed reagents, Drafting and revising the article; DBF, EAB, Conception and design, Contributed reagents, Drafting and revising the article; HEG, MAN, Conception and design, Analysis and interpretation of data, Drafting and revising the article

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
