## [Decision Letter]

Thank you for submitting your work entitled "Resolving Dual Binding Modes of Cellulosome Cohesin-Dockerin Complexes using Single-Molecule Force Spectroscopy" for peer review at *eLife*. Your submission has been favorably evaluated by John Kuriyan (Senior Editor), a Reviewing Editor, and two reviewers. One of the two reviewers, Jie Yan, has agreed to reveal his identity.

The reviewers have discussed the reviews with one another and the Reviewing Editor has drafted this decision to help you prepare a revised submission.

Summary:

The manuscript by Jobst et al. describes the using of site-directed mutagenesis, protein engineering, and atomic force microscopy to assess the dual mode of binding of the cellulosomal type-I dockerin module to its cognate scaffoldin type-I cohesin module.

The cellulosome is a highly efficient lignocellulosic degrading complex produced by several anaerobic bacteria. The high-affinity type-I cohesin-dockerin interaction mediates the assembly of enzymatic subunits onto the central scaffoldin subunit. X-ray crystallographic studies have revealed that type-I dockerin modules, through its internal sequential symmetry, can display two modes of binding on the surface of the type-I cohesin module. These studies have been complemented with biophysical studies (e.g. isothermal titration calorimetry) and have involved the use of dockerin mutants to distinguish the two binding modes. Direct observation of this dual binding mode in solution has remained elusive, including assessing any preference for a particular mode/orientation.

The authors observed either a single force peak or two force peaks that are separated by ~ 8 nm preceding the final dissociation of the complex, which led to their critical hypothesis that they may correspond to the dual-binding modes of Coh-Doc complexes. The authors excluded the possibilities that these characteristic force peaks were from unfolding of CBM, xylanase, and partial unfolding of cohesion by various strictly designed control experiments. By additional studies using (GS)8-insert located between helix 1 and 2, the authors were finally able to determine that the double-peak of unfolding force separated by 8-nm is from unfolding involving loop 1 and helix 1 at the N-terminus of dockerin.

Overall, this is an important and solid study that provides important insights to the mechanism of Coh-Doc interaction. This manuscript is clearly and concisely written. The methods used are highly appropriate and the analysis and interpretation of the data appears sound. Indeed, the work represents a novel and revealing study. One weakness of the manuscript is that there is no clear exposition on if and why the two binding modes may have physiological consequences.

Essential revisions:

1) Some additional information in the Introduction to highlight the biological and biophysical significance of the cellulosome cohesin-dockerin interactions and in the Discussion to show the benefits of having two binding modes should be added. Otherwise, the readers may feel that the two binding modes may be of peculiar novelty that may be of no physiological consequences.

2) The authors stated that the binding of Doc helix 1 to Coh corresponds to the binding mode A, while that of Doc helix 3 to Coh corresponds to the binding mode B. It will be helpful to indicate all interaction bonds at the binding interface in Figure 1 or in a new figure panel for each binding mode. It will also be helpful to add an arrow to the N-terminus of Doc in Figure 1 to indicate location where the force is applied. With such information illustrated in the figure, it will be easier for readers to understand how force is applied, and why the mode A may have double force peaks prior to rupture.

3) The authors proposed that in the mode A the first barrier of the double peaks represents unfolding of the N-terminal calcium binding loop and unraveling of alpha helix 1 up to the Lys-Arg pair at sequence positions 18 and 19. But my impression is that in the mode A, the binding is through the interaction between Doc helix 1 and Coh. Do the authors suggest that unfolding of the N-terminal calcium-binding loop always precedes the dissociation of the Doc helix 1 from Coh? Does the N-terminal calcium-binding loop also bind to Coh? If not, how can its unfolding be thought of as the first barrier against rupture? What will happen if Ca^2+^ is removed from solution? Some clarifications and additional discussions will be helpful.

4) In their Q3 mutant, the binding mode B is excluded. Should we expect to see predominant double force peak preceding rupture since only the mode A is possible? Why do we still see 41% of single force peak, similar to that from the wild type? Maybe we misunderstood something. If we did, please clarify in the reply letter and in a revised manuscript.

---

## [Author Response]

*Essential revisions:*

*1) Some additional information in the Introduction to highlight the biological and biophysical significance of the cellulosome cohesin-dockerin interactions and in the Discussion to show the benefits of having two binding modes should be added. Otherwise, the readers may feel that the two binding modes may be of peculiar novelty that may be of no physiological consequences.*

We thank the reviewers for the suggestion to highlight the biological and biophysical significance of Coh-Doc interactions, and the benefits of having a dual binding mode.

In our prior draft, we stated in the Introduction: “The dual binding mode is thought to increase the conformational space available to densely packed enzymes on protein scaffolds, and to facilitate substrate recognition by catalytic domains within cellulosomal networks (Bayer et al., 2004). From an evolutionary perspective, the dual binding mode confers robustness against loss-of-function mutations, while allowing mutations within Doc to explore inter-bacterial species cohesin-binding promiscuity in cellulosome-producing microbial communities.” We added to this: “Coh:Doc interactions and dual binding modes are therefore important in the context of cellulose degradation by cellulosome-producing anaerobic bacterial communities.”

We have also now added the following text to the Discussion section: “The biological significance of Coh-Doc interactions in the context of cellulosome assembly and catalysis cannot be overstated […] the single-molecule biophysical approach presented here based on differences in mechanical hierarchies will facilitate further study into the significance of the dual binding mode.”

*2) The authors stated that the binding of Doc helix 1 to Coh corresponds to the binding mode A, while that of Doc helix 3 to Coh corresponds to the binding mode B. It will be helpful to indicate all interaction bonds at the binding interface in Figure 1 or in a new figure panel for each binding mode. It will also be helpful to add an arrow to the N-terminus of Doc in Figure 1 to indicate location where the force is applied. With such information illustrated in the figure, it will be easier for readers to understand how force is applied, and why the mode A may have double force peaks prior to rupture.*

We added panel D to Figure 1, which now shows a close-up of the interface together with the locations and directions of the applied forces on Doc and the Doc residues dominating both binding modes. The figure caption was extended with “(D) Close-up view of the interface for each binding mode with arrows indicating the location and direction of applied force. Binding residues 11, 12, 18 and 19 for binding mode A and 45, 46, 52 and 53 for binding mode B are shown as blue stick models. The Coh domain is oriented the exact same way in both views.”

*3) The authors proposed that in the mode A the first barrier of the double peaks represents unfolding of the N-terminal calcium binding loop and unraveling of alpha helix 1 up to the Lys-Arg pair at sequence positions 18 and 19. But my impression is that in the mode A, the binding is through the interaction between Doc helix 1 and Coh. Do the authors suggest that unfolding of the N-terminal calcium-binding loop always precedes the dissociation of the Doc helix 1 from Coh?*

The answer is yes for double peaks in binding mode A. That is the scenario that is consistent with the 8 nm contour length increment and the behavior of the mutants.

Does the N-terminal calcium-binding loop also bind to Coh?

Yes, a portion toward the c-terminal side of the n-terminal calcium loop is involved in Coh binding (i.e., Ser-Thr pair at position 11-12).

*If not, how can its unfolding be thought of as the first barrier against rupture?*

The calcium-binding loop is involved in Coh binding as the reviewers suggested. The binding residues (S11-T12) are highly conserved among Doc domains, and form hydrogen bonds to Coh. In our prior work (Stahl et al., PNAS, 2012), we mutated the Coh residue involved in hydrogen bonding to this site (Coh D39A mutant), and saw huge decrease in rupture forces. We also speculate that an intramolecular clasp, which involves hydrophobic stacking between the N- and C-termini of Doc contributes to the 1^st^ barrier. We are currently working with collaborators on steered molecular dynamics simulations to determine which of these effects is dominant.

*What will happen if Ca^2+^ is removed from solution? Some clarifications and additional discussions will be helpful.*

We refer the reviewers here to our prior work (Stahl et al., PNAS, 2012), where we performed extensive force-dependent calcium removal experiments. We found calcium can be removed from Doc under applied force, and when that happens it loses its binding ability. Interestingly, EDTA at low mM concentrations is not sufficient to remove Ca from bound Doc domains, but under force this loss of Ca is apparent and results in loss of binding function.

To address these comments, we have added the following additional discussion to the manuscript: “A portion of the N-terminal calcium binding loop (i.e., residues S11-T12) is involved in binding to D39 in Coh. The first peak of the double events is attributed to breakage of this interaction and simultaneous unfolding of calcium loop 1 and alpha helix 1 up to the Lys-Arg pair at sequence positions 18 and 19.”

4) In their Q3 mutant, the binding mode B is excluded. Should we expect to see predominant double force peak preceding rupture since only the mode A is possible? Why do we still see 41% of single force peak, similar to that from the wild type? Maybe we misunderstood something. If we did, please clarify in the reply letter and in a revised manuscript.

We were surprised too with this result. It would have been wonderful if the Q3 mutant showed only double peaks, but 41% were still singles. We looked further into this and found that even though the simple double vs. single classification seemed to indicate no differences between wild-type and Q3, when we quantified the rupture force differences between double peak events, a significant difference became apparent. We quantified the relative difference in rupture force between the two peaks in the double events (Figure 4). We saw that for Q3-mutant doubles, the second peak is lower in force than the first (Figure 4, bottom) while the wild-type showed behavior where the second peak was higher (Figure 4). Our interpretation is that since the second barrier is weaker than the first for the Q3 mutant, sometimes the first is missed completely (i.e., the molecule jumps over both barriers at once). Generally, each binding mode still allows for the occurrence of a single event, in which the whole Doc domain unbinds without additional unfolding, regardless of the mode the complex is currently in.

Also, measuring doubles for the Q3 mutants presents a technical challenge because the double event second peaks can be very small due to the lowered forces, and difficult to identify. It is therefore possible that the double peaks for Q3 are slightly undercounted.

To clarify this point, we have added the following to the Results section: “This weaker 2^nd^ double peak for the Q3 mutant combined with similar single/double peak ratios as wild-type leads us to believe that the number of double peaks is being underestimated systematically for the Q3 mutant. Generally, each binding mode still allows for the occurrence of a single event, in which the whole Doc domain unbinds without additional unfolding. Since the second barrier is weaker than the first for the Q3 mutant, the probability for the molecule to pass both barriers simultaneously is increased, thus resulting in a high percentage of single events.”